# Serum Insufficiency Induces RANKL-Independent Osteoclast Formation during Developing Ischemic ONFH

**DOI:** 10.3390/biomedicines9060685

**Published:** 2021-06-17

**Authors:** Yi-min Hsiao, Chih-Chien Hu, Mei-Feng Chen, Chih-Hsiang Chang, Yu-Tien Chiu, Yuhan Chang

**Affiliations:** 1Department of Orthopedic Surgery, Chang Gung Memorial Hospital, Taoyuan 33305, Taiwan; skyyimin@gmail.com (Y.-m.H.); r52906154@cgmh.org.tw (C.-C.H.); ccc0810.chang@gmail.com (C.-H.C.); 2Bone and Joint Research Center, Chang Gung Memorial Hospital, Taoyuan 33305, Taiwan; mfchen0@gmail.com (M.-F.C.); mimiko690@gmail.com (Y.-T.C.); 3College of Medicine, Chang Gung University, Taoyuan 33302, Taiwan; 4Graduate Institute of Clinical Medical Sciences, College of Medicine, Chang Gung University, Taoyuan 33302, Taiwan

**Keywords:** ischemia, osteonecrosis, serum insufficiency, osteogenesis, osteoclast

## Abstract

Blood supply interruption induces hypoxia and reduces serum provision to cause ischemia-induced osteonecrosis, including avascular osteonecrosis of the femoral head (ONFH). Oxygen deficiency (hypoxia) is known to induce different expression patterns in osteoblasts and osteoclasts, which have been extensively studied. However, the effects of serum insufficiency in nutrients, growth factors, and hormones on osteoblast and osteoclast activity in the damaged area and nearby regions remain poorly understood. In this study, the expression of osteoblast and osteoclast marker proteins was elucidated through in vitro and ex vivo studies. The results indicate that serum insufficiency accelerates the formation of monocyte-derived osteoclasts. The combined effect of serum insufficiency and hypoxia (mimicking ischemia) suppressed the activity of alkaline phosphatase and calcification in osteoblasts after the stimulation of osteogenic growth factors. Serum insufficiency increased the activity of tartrate-resistant acid phosphatase, expression of phosphorylated extracellular signal-regulated kinases, and production of reactive oxygen species in monocyte-derived osteoclasts in the absence of receptor activator of nuclear factor kappa-Β ligand stimulation. The findings indicate that changes in the expression of osteoblast and osteoclast markers in necrotic bone extracts were similar to those observed during an in vitro study. These results also suggest that serum insufficiency may be involved in the regulation of osteoclast formation in patients with ONFH.

## 1. Introduction

Osteonecrosis of the femoral head (ONFH) is a pathological condition of the hip joint that is caused by disruption of the blood supply to the femoral head [1,2]. Symptoms of soft tissue necrosis, including cell swelling and inflammatory responses, do not occur in ONFH, and the exact mechanism of this ischemic disease is unknown [3,4]. The interruption in blood supply that is a pathophysiological characteristic of this disease induces ischemia (serum (nutrient) insufficiency and hypoxia), which disrupts the balance of osteoblast and osteoclast activity at the damaged area of the femoral head during ONFH progression [5,6,7].

Hypoxia is known to promote osteoclast differentiation and suppress osteoblast activity through receptor activator of nuclear factor kappa-Β ligand (RANKL) stimulation [8,9], and RANKL induces the activation of downstream factors, including reactive oxygen species (ROS), phosphorylated extracellular signal-regulated kinase (ERK), and nuclear factor of activated T-cells cytoplasmic 1 (NFATc1). Consequently, the expression of downstream targets, including tartrate-resistant acid phosphatase (TRAP, an enzyme highly expressed in osteoclasts and thought to participate in osteoclast-mediated bone turnover) and cathepsin K (CTSK, the major collagenolytic cysteine proteinase expressed in osteoclasts), is activated during osteoclast formation [10]. Clinical treatments for ONFH include the application of core decompression and the use of osteoclast inhibitors for reducing hypoxia and suppressing osteoclast activity, respectively; however, no treatment can efficiently inhibit or fully repair bone loss [11,12,13,14,15]. Several studies have reported that the combination of core decompression with stem cells or osteoclast inhibitors is more effective than other methods for treating ONFH [16,17,18].

In fact, interruption of the supply of nutrients from serum (blood) flow in the necrotic area and nearby regions during ONFH development has also been observed [19]. However, the effects of serum insufficiency on osteoblast and osteoclast activity in cases of serum insufficiency and ischemic conditions remain poorly understood. In this study, we investigated pattern changes in osteoblasts and osteoclasts with serum insufficiency and identified the effects of serum insufficiency and ischemic conditions on osteoblast and osteoclast activity in vitro. We also provide ex vivo evidence of changes in osteoblast and osteoclast activity under ischemic stress to identify the possible mechanism of ONFH development.

## 2. Results

### 2.1. Serum Insufficiency Suppressed Proliferation of Monocytes but Induced Osteoclastogenesis

To elucidate the effects of serum insufficiency, hypoxia, and ischemia on the formation of osteoclasts, RAW264.7 cells (mouse monocytes) were used for the experiments. The viability of RAW264.7 cells was suppressed following treatments to achieve serum insufficiency and cobalt chloride (CoCl_2_)-induced hypoxia (200 and 400 µM) compared with the viability of cells subjected to a vehicle control (Appendix A). Because 400 µM CoCl_2_ induced cell death (70%–80%) on Day 5, 200 µM CoCl_2_ was used in subsequent experiments (Appendix A). Preosteoclast-like cells (irregularly shaped) were observed under conditions of CoCl_2_-induced hypoxia, serum insufficiency, and ischemia on Day 3 but not under the control conditions (round monocytes); both hypoxia and serum insufficiency reduced the cell density and increased the number of irregularly shaped cells until Day 5 (Appendix A).

To determine whether serum insufficiency induced preosteoclast and osteoclast formation, TRAP expression and activity were detected after treatments on Days 3, 5, and 7. Compared with the control group (10% fetal bovine serum (FBS) alone; few TRAP-positive mononuclear cells and fused polykaryons were detected), serum insufficiency alone and in combination with 200 µM CoCl_2_-induced hypoxia resulted in more TRAP-positive mononuclear cells and fused polykaryons; this observation was similar to the results of RANKL stimulation (Figure 1A–C). Serum insufficiency alone and in combination with hypoxia (the ischemic condition) significantly increased the number of TRAP-positive osteoclasts (≥3 nuclei) on Day 3; however, hypoxia significantly increased the number of TRAP-positive osteoclasts from Day 5 onward (Figure 1D). These results indicate that in a manner similar to RANKL stimulation, serum insufficiency facilitated osteoclast formation and fusion. Notably, treatments to induce serum insufficiency alone and in combination with hypoxia resulted in stronger TRAP activity than that observed in the 10% FBS treatment group on Day 3 (Figure 1E). Additionally, similar to the results of RANKL stimulation, serum insufficiency induced the production of ROS, and ischemia continuously increased the number of ROS on Day 7 (Figure 1F).

According to the aforementioned results, the expression of osteoclast markers (TRAP, CTSK, and NFATc1, and phosphorylated ERK1/2 (pERK1/2)) was detected after the administration of relevant treatments (Figure 2A). The TRAP levels did not significantly differ between the serum-rich groups (the group with induced hypoxia and that subjected to RANKL stimulation) until Day 7; however, the ischemic condition dramatically increased the TRAP levels on Days 3 and 7 (Figure 2B). In contrast to the control group (10% FBS treatment alone), the serum insufficiency (0.1% FBS) and ischemia groups did not exhibit increases in CTSK or NFATc1 expression; only RANKL stimulation increased the expression levels of both proteins (Figure 2C,D). Notably, the pERK1/2 level was increased under the 0.1% FBS condition (Figure 2E). The ischemic condition increased TRAP levels but slightly suppressed the phosphorylation of ERK, in contrast to the conditions of 0.1% FBS alone and in combination with RANKL stimulation. Taken together, these results indicate that increases in osteoclast formation due to serum insufficiency and ischemia may be related to the activation of the pERK pathway and TRAP activity in the absence of RANKL stimulation.

### 2.2. Serum Reperfusion Inhibited Serum Insufficiency-Induced Osteoclastogenesis

To elucidate whether serum reperfusion affects osteoclast formation induced by serum insufficiency or ischemia, monocytes were cultured in 0.1% FBS with or without hypoxia for 3 days; the culture medium was then changed to a medium containing 10% FBS for 4 days to determine the number of TRAP-positive osteoclasts, evaluate the production of ROS, and determine the pERK expression level. Increases in the number of TRAP-positive osteoclasts, which were influenced by serum insufficiency, ischemia, or RANKL stimulation, were suppressed after serum reperfusion from Days 4 to 7 (Figure 3A). Moreover, serum reperfusion reduced ischemia-induced ROS production but mitigated the increase in pERK expression (Figure 3B,C). Taken together, these results indicate that the regulation of ROS production and ERK phosphorylation under conditions of serum insufficiency and ischemia may be related to the formation of monocyte-derived osteoclasts in the absence of RANKL stimulation.

### 2.3. Ischemia Suppressed Calcification, Alkaline Phosphatase Activity, and the Activation of Osteoblasts after Stimulation of Osteogenic Growth Factors

Serum insufficiency accelerated osteoclast formation, but the effect of serum insufficiency on osteoblast differentiation requires further investigation. There are several markers of osteoblast differentiation, including calcification (a marker of the late stage) and high levels of alkaline phosphatase (ALP, a marker of the early stage), which were first detected after the induction by osteogenic growth factors (OGFs). OGF-induced MC3T3-E1 cells were used to analyze calcification and ALP activity in situations of stress due to serum insufficiency or hypoxic or ischemic conditions.

In comparison with the control group under the 10% FBS condition, in which calcified osteoblasts (red nodules) were not observed on Days 14, 21, and 28, the OGF-induced group exhibited calcified osteoblasts only on Days 21 and 28 (Figure 4B). The formation of red nodules was not detected after treatment to induce a hypoxic condition, serum insufficiency, or the ischemic condition until Day 28 (Figure 4C,E,F). The calcium concentration was only increased after OGF stimulation alone on Days 21 and 28; the calcium concentration in the other treatment groups was not detectable (Figure 4G). Notably, prolonged treatment to induce hypoxia alone and in combination with serum insufficiency triggered neuron-like differentiation from Days 21 to 28 (Figure 4C,F). Additionally, ALP activity was detected in the early stage of osteoblast differentiation. The results reveal that ALP expression was slightly increased in the 10% FBS group (basal levels) during culturing (Figure 5A). In contrast to the results for the 10% FBS group, ALP activity was dramatically increased (at least twofold) from Days 3 to 7 after OGF stimulation (Figure 5B) and was inhibited under the conditions of hypoxia, serum insufficiency, and ischemia (Figure 5C–F). Moreover, the ischemic condition was more effective in suppressing ALP activity after OGF stimulation (Figure 5G).

In addition to these histochemical and enzymological results, the expression levels of runt-related transcription factor 2 (RUNX2, a preosteoblast marker) and its downstream target osteopontin (OPN, an osteoblast marker) [20] were detected through immunoblotting after the application of appropriate treatments (Figure 6A). The RUNX2 expression induced by OGFs decreased on Days 3 and 7 under the hypoxic condition but not under the condition of 10% FBS treatment alone; however, 0.1% FBS treatment increased RUNX2 expression levels on Day 3, but this RUNX2 increase was undetected on Day 7 (Figure 6B). In contrast to the findings on RUNX2 expression, 0.1% FBS treatment had no significant effect on OPN expression on Day 3 but significantly increased the expression level on Day 7 after the induction of OGFs. OPN expression was considerably inhibited by CoCl_2_ treatment, regardless of whether the culture medium contained 10% or 0.1% FBS (Figure 6C). In addition, serum reperfusion did not rescue ALP activity (Figure 6D).

### 2.4. Ischemia-Induced Pattern Changes in Osteoblast and Osteoclast Proteins in Patients with and without Osteonecrosis

Micro-computed tomography (µCT) images were used for detecting morphological differences between the control and ONFH groups; bone loss occurred in the ONFH groups (Figure 7A). The total protein content was extracted from control (C), non-necrotic (N), and necrotic (A) bone tissues for immunoblotting. Regarding the osteoblast markers, RUNX2, ALP, and OPN levels were lower in necrotic regions than in the control group (Figure 7B). Regarding osteoclast markers, TRAP and CTSK exhibited low expression levels and were undetected in the control group, respectively. In comparison with the TRAP and pERK1/2 expression levels in the control group, those in non-necrotic and necrotic tissues were significantly higher. Moreover, the expression levels of CTSK were higher in non-necrotic tissues than in the control group (Figure 7B). A graphic abstract of the possible mechanism of serum insufficiency and ischemia is presented in Appendix A.

## 3. Discussion

This study combined the results of in vitro culturing of osteoblast and osteoclast precursors to elucidate the effects of serum insufficiency alone and in combination with ischemia in disrupting the balance of osteoblast and osteoclast formation, as observed in patients with ONFH. Serum insufficiency increased the formation of monocyte-derived osteoclasts in vitro, regardless of whether serum insufficiency occurred in combination with hypoxia. Our findings may suggest that serum insufficiency accelerates osteoclastogenesis without RANKL stimulation or the inhibition of osteoblast activity under ischemic conditions.

For osteoclastogenesis, the decrease in cell density observed under conditions of serum insufficiency and ischemia (without RANKL stimulation) may be attributable to the acceleration of preosteoclast fusion (Figure 1) or the possible inhibition of osteoclast formation after serum reperfusion reduced the number of osteoclasts (Figure 3). The expression levels of TRAP and pERK1/2 increased substantially under conditions of serum insufficiency in vitro representing the conditions of ONFH tissues (Figure 2, Figure 3, and Figure 7). CTSK exhibited low expression levels under conditions of serum insufficiency, hypoxia, and ischemia. The results indicate that lower levels of CTSK expression are associated with stronger TRAP activity (Figure 2B,C). We propose that TRAP and CTSK expression play different roles in regulating osteoclast differentiation or activation. Increased levels of TRAP expression have been reported to be attributable to increases in the number of osteoclasts in mice [21,22]. The expression of NFATc1, an upstream transcription factor for TRAP and CTSK, was not detected in the groups with serum insufficiency, hypoxia, or ischemia (Figure 2D) or in bone extracts (data not shown). The pattern of NFATc1 expression was similar to that of CTSK expression under different treatments; this suggests that CTSK expression may be closely related to NFATc1 expression through the RANKL-dependent pathway. However, this result also suggests that serum insufficiency may regulate TRAP activity through the RANKL-independent pathway.

ROS and oxidative stress are known to be responsible for the development of ROS-mediated orthopedic diseases, including osteopenia and postmenopausal osteoporosis [23,24,25,26,27,28]. This explains the increase in ROS under the conditions of serum insufficiency and ischemia, which was similar to the increase in ROS observed in the RANKL-stimulated group on Day 7 (Figure 1F) that was suppressed after serum reperfusion (Figure 3B). The same phenomenon was also observed in the expression of pERK (Figure 3C). Therefore, we hypothesize that serum-insufficiency-induced intracellular ROS and pERK activation may be related to the regulation of TRAP levels and osteoclast formation without RANKL stimulation.

Our results show that lower serum concentrations suppressed calcification, regardless of whether hypoxia also occurred (Figure 4), and increased RUNX2 expression levels within the first 3 days (Figure 6). Therefore, ALP and OPN activity exerted no effect on protein levels on Day 3 (Figure 5 and Figure 6). These results suggest that serum insufficiency may promote cell cycle arrest in the G0/G1 phase to inhibit osteoblast activity, which is in line with reports on primary osteoblasts [29,30]. In addition, the neuron-like differentiation observed after prolonged hypoxic and ischemic treatments may be attributable to hypoxia rather than to serum insufficiency (Figure 4). The serum-insufficiency-induced suppression of ALP activity was not rescued after serum reperfusion (Figure 6), suggesting that osteoblast activity may not be rescued after tissue damage.

In summary, we first observed that serum insufficiency plays a prominent role in the differentiation of monocyte precursors and the cell fusion of preosteoclasts. Although decreases in serum concentrations activated the pERK pathway, the exact mechanism of osteoclastogenesis induced by serum insufficiency or ischemia still requires elucidation. Our findings provide new insight into the effect of serum insufficiency through the induction of ROS synthesis and activation of pERK expression on the acceleration of osteoclast formation in ischemia-induced osteonecrosis.

## 4. Materials and Methods

### 4.1. Cell Culture and Treatments

MC3T3-E1 cells (ATCC CRL-2593; American Type Culture Collection, VA, USA) and RAW264.7 cells (ATCC TIB-71; American Type Culture Collection) were cultured in alpha Eagle’s minimum essential medium (Thermo Fisher Scientific, Waltham, MA, USA) containing 10% FBS (Thermo Fisher Scientific) and 1% penicillin and streptomycin (Thermo Fisher Scientific, Waltham, MA, USA). The hypoxic condition was induced with 200 and 400 µM cobalt (II) chloride (CoCl_2_; Sigma-Aldrich, St. Louis, MO, USA) [31]. To induce serum insufficiency, cells were cultured in a medium containing 0.1% FBS. The ischemic condition was induced through the use of 0.1% FBS and cobalt-induced hypoxia.

### 4.2. Protein Extraction and Immunoblotting Analysis

Extraction buffer (radioimmunoprecipitation assay buffer plus 1× proteinase and phosphatase inhibitors; Sigma-Aldrich, St. Louis, MO, USA) was used to extract the total protein content from 200 mg powders or 1 × 10^6^ cells. After the sample was placed in 10% resolving 1× sodium dodecyl sulfate–polyacrylamide gel, the proteins were transferred to a nylon membrane. Blocking was achieved by placing the nylon membrane in 5% skimmed milk in 1× phosphate-buffered saline containing 0.05% Tween 20 (1× PBST); subsequently, primary and secondary antibodies were diluted in 2% skimmed milk in 1× PBST (Appendix A) for protein detection. Signals were developed by using electrochemiluminescence solution (PerkinElmer, Waltham, MA, USA) and films (GE Healthcare, Chicago, IL, USA).

### 4.3. ALP Activity and Calcification

Cells were seeded onto a 12-well plate, with 1 × 10^4^ MC3T3-E1 cells per well. OGFs (100 nM dexamethasone, 0.05 mM ascorbic acid 2-phosphate, and 10 mM b-glycerophosphate; Sigma-Aldrich, St. Louis, MO, USA) were added to trigger osteoblast differentiation under different treatments. After the cells had been treated, they were fixed in 60% citrate-buffered acetone and then incubated with a mixture of Fast Violet B Salt and naphthol AS-MX phosphate alkaline solutions (Sigma-Aldrich, St. Louis, MO, USA) for the detection of ALP expression. ALP activity and the total number of cells were measured using the TRACP & ALP assay kit (TakaRa Bio, Kusatasu, Shiga, Japan) and water-soluble tetrazolium salt (WST-1) reagent (Roche, Basel, Switzerland), respectively, in accordance with the manufacturers’ instructions. Cells were fixed with 10% paraformaldehyde after treatment, and then a 2% Alizarin Red S solution (ScienCell, Carlsbad, CA, USA) was used for detecting calcification. Calcium LiquiColor Assay (Stanbio Laboratory, Boeme, TX, USA) and WST-1 reagent (Roche, Basel, Switzerland) were respectively used in accordance with the manufacturers’ instructions to determine the calcium content and number of cells after treatment.

### 4.4. TRAP Staining and Activity

For the detection of TRAP expression, 1 × 10^4^ RAW264.7 cells were seeded into each well in a 3-well cell culture slide (Thermo Fisher Scientific, Waltham, MA, USA). To maintain a stable condition, 75% of the medium was changed every 2–3 days. After 90% confluency was achieved, TRAP staining and quantification of TRAP activity were performed using a TRACP & ALP double-stain kit and TRACP & ALP assay kit (TakaRa Bio, Kusatasu, Shiga, Japan) in accordance with the manufacturer’s instructions.

### 4.5. ROS Assay

Cells were seeded onto a 96-well dish, with 5 × 10^3^ RAW264.7 cells per well. After 90% confluency was achieved and appropriate treatments had been applied, ROS detection was performed on Days 3 and 7 with CM-H2DCFDA (Thermo Fisher Scientific, Waltham, MA, USA) in accordance with the manufacturer’s instructions.

### 4.6. Sample Collection from Patients

This study was approved by our institutional review board and complied with current ethical standards (IRB Number: 201901555B0). All patients met the criteria for radiological diagnosis and were scheduled to undergo total hip arthroplasty during the period from November 2019 to January 2021. Femoral heads were collected from seven patients (five with femoral neck fractures and two with osteoarthritis of the hip) to serve as the control groups (aged 67–80 years) and from seven patients (with idiopathic or alcohol-related ONFH) to serve as the experimental groups (aged 48–76 years; Ficat stages III–IV).

### 4.7. Image Analysis

Imaging of the femoral heads was performed using a SkyScan 1272 µCT scanner (Bruker, Kontich, Belgium). Images of cell cultures were captured using an Olympus CKX41 microscope (Olympus Co., Tokyo, Japan) connected to a Canon EOS 800D camera (Canon Inc., Tokyo, Japan). Films of immunoblotting and 12-well dishes were scanned using a CanoScan LiDE 400 (Canon Inc., Tokyo, Japan). TRAP images were captured using Leica DFC7000 T Microsystems (Leica, Wetzlar, Germany). All images were edited in Photoshop CS3 (Adobe Inc., San Jose, CA, USA) and ImageJ version 2.0 (https://imagej.net/software/imagej2/, accessed on 17 June 2021).

### 4.8. Statistical Analysis

All data consisted of at least three biological replicates. Quantitative results are presented as the mean ± standard deviation, and statistical analysis was performed in GraphPad Prism software, version 6.0 (GraphPad Software, San Diego, CA, USA). The results were analyzed through two-way analysis of variance with Tukey’s honestly significant difference test and the paired Student’s *t*-test (Figure 7). The 95% confidence level was set according to Tukey’s honestly significant difference test, and a *p* value of less than 0.05 was considered statistically significant in this study.

## Figures and Tables

**Figure 1 biomedicines-09-00685-f001:**
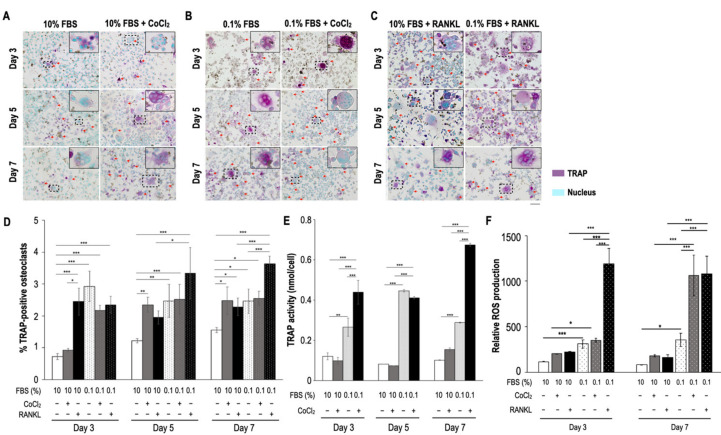
Monocyte TRAP activity and osteoclast formation after treatment with serum insufficiency with or without hypoxia. Serum insufficiency was induced with 0.1% FBS, and a hypoxic condition was induced with 200 µM cobalt chloride. Purple indicates TRAP expression, and cyan indicates nuclei. The upper right images are magnifications of the dashed box region in each panel. Arrows indicate fused polykaryons (≥2 nuclei in single cells). (**A**) Formation of TRAP-positive cells under hypoxic conditions. (**B**) Effect of ischemia on the formation of TRAP-positive cells. (**C**) Effect of RANKL stimulation on cells treated with 10% and 0.1% FBS conditions. Cells that underwent RANKL stimulation were the positive controls. (**D**) Number of TRAP-positive osteoclasts. Bar height indicates the number of TRAP-positive osteoclasts (≥3 nuclei), which were counted in five random areas in each well and normalized to the total cell number. (**E**) Quantitative analysis of TRAP activity. The optical density (O.D.) value of the final products of the TRAP substrates was normalized to that of the cell lysates (the internal control). All data are presented as mean ± SD (*n* = 3). (**F**) ROS production after treatments. The O.D. values of different groups were normalized to the total protein content. Each bar corresponds to a single group and represents the mean ± SD (*n* = 6). Asterisks at the tops of bars indicate significant differences between two groups (* *p* < 0.05, ** *p* < 0.01, and *** *p* < 0.001) in (**DF**). Abbreviations: FBS, fetal bovine serum; RANKL, receptor activator of nuclear factor kappa-Β ligand; ROS, reactive oxygen species; SD, standard deviation; TRAP, tartrate-resistant acid phosphatase. Scale bar: 50 µm.

**Figure 2 biomedicines-09-00685-f002:**
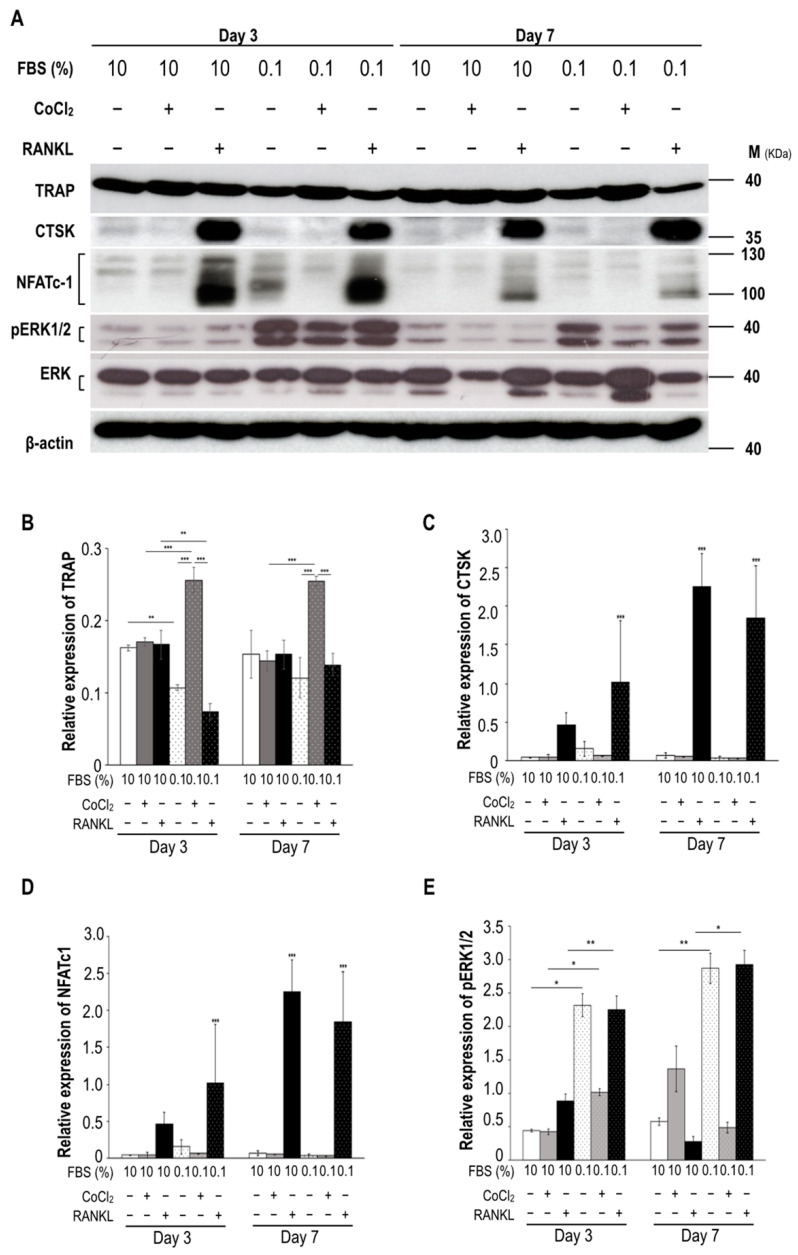
Expression of osteoclast markers subject to stress. (**A**) Immunoblotting was used to detect marker proteins in RAW264.7 lysates. Antibodies of TRAP, CTSK, NFATc1, and pERK1/2 were used to detect osteoclast expression after different treatments. Quantification results are also displayed for (**B**) TRAP, (**C**) CTSK, (**D**) NFATc1, and (**E**) pERK1/2. The identities of the level of TRAP, CTSK, and NFATc1 proteins were normalized to the level of β-actin; pERK1/2 was normalized to the total ERK1/2 protein level. Brackets indicate isoforms of NFATc1 (from 90 to 140 KDa) and ERK1/2 (44 and 42 KDa) in RAW264.7 cells. The identity of NFATc-1 is the combination of three isoforms. All data are presented as the mean ± SD (*n* = 3); asterisks at the tops of the bars indicate significant differences (* *p* < 0.05, ** *p* < 0.01, *** *p* < 0.001). Abbreviations: FBS, fetal bovine serum; CTSK, cathepsin K; NFATc1, nuclear factor of activated T-cells cytoplasmic 1; pERK1/2, phosphorylated extracellular-regulated kinase 1/2; TRAP, tartrate-resistant acid phosphatase.

**Figure 3 biomedicines-09-00685-f003:**
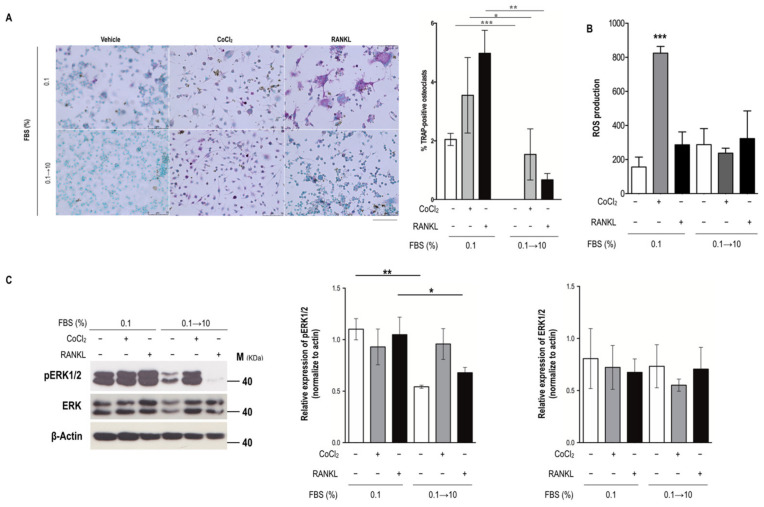
Number of TRAP-positive osteoclasts and ROS production level after serum reperfusion. Purple indicates TRAP expression, and cyan indicates nuclei. (**A**–**C**) After the application of 0.1% FBS or the ischemic condition treatment for 3 days, the medium was replaced with a medium containing 10% FBS for 4 days. All data are presented as mean ± SD (*n* = 3). (**A**) Osteoclast formation. Bar height indicates the number of TRAP-positive osteoclasts (≥3 nuclei) counted in five randomly selected areas in each well, normalized to the total cell number. (**B**) ROS production. Bar height indicates the fluorescence intensity after incubation with 2′,7′-dichlorodihydrofluorescein diacetate (DCFDA). (**C**) Expression of pERK. Bar height indicates the identity of pERK, normalized to that of total ERK protein. Asterisks at the tops of bars indicate significant differences between two groups (* *p* < 0.05, ** *p* < 0.01, and *** *p* < 0.001). Abbreviations: FBS, fetal bovine serum; RANKL, receptor activator of nuclear factor kappa-Β ligand; ROS, reactive oxygen species; TRAP, tartrate-resistant acid phosphatase. Scale bar: 50 µm.

**Figure 4 biomedicines-09-00685-f004:**
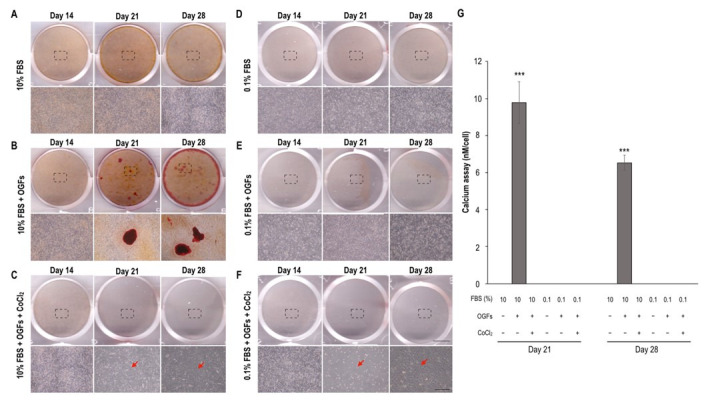
Calcification of OGF-induced MC3T3-E1 cells subjected to the stress of serum insufficiency with or without hypoxia. Red indicates calcification detected through Alizarin Red staining. Calcification was detected on Days 14, 21, and 28 after the administration of appropriate treatments. Images of 12-well dishes were captured with a scanner, and cell images are magnifications of the dashed box region. (**A**) Incubation with 10% FBS alone (normal control) did not trigger calcification until Day 28. (**B**) OGF-induced cells (positive control). Red nodules on the wells indicate calcification. (**C**) Effect of hypoxia on the calcification of OGF-induced cells. Arrows indicate neuron-like cells. (**D**) Effect of serum deprivation on pre-osteoblasts. (**E**) Coincubation with OGFs and 0.1% FBS. (**F**) Effect of ischemia on calcification. (**G**) Calcification assay. The O.D. value of the calcium concentration in each group was normalized to the total cell number. Each bar corresponds to a single group and represents the mean ± SD (*n* = 3). Asterisks indicate significant differences (*** *p* < 0.001). Abbreviations: Ctrl, control; FBS, fetal bovine serum; OGFs, osteogenic growth factors.

**Figure 5 biomedicines-09-00685-f005:**
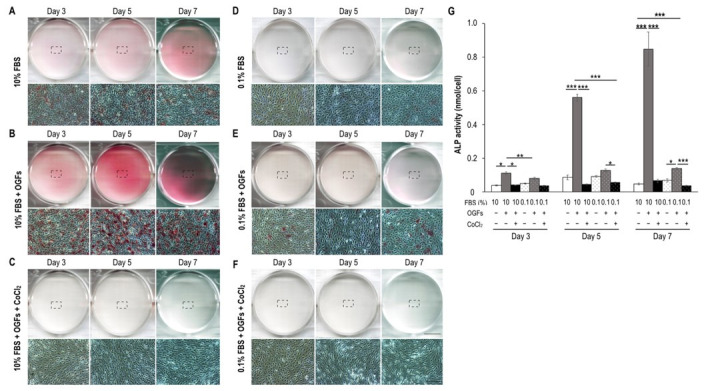
ALP activity in OGF-induced pre-osteoblasts subjected to conditions of serum insufficiency with and without hypoxia. Red indicates ALP expression in the wells. Cell images are magnifications of the dashed box regions. (**A**) ALP expression of MC3T3-E1 cells under the 10% FBS condition. Lower ALP expression was only observed for 10% FBS alone during the culturing period when it served as a normal control. (**B**) OGF-induced cells. OGFs considerably increased ALP expression when they were serving as a positive control. (**C**) Cotreatment with CoCl_2_ and OGFs. Treatment with 200 µM CoCl_2_ was applied to mimic hypoxic conditions, and it suppressed ALP expression in OGF-induced cells. (**D**) Effect of serum deprivation alone. Serum deprivation was induced through incubation with 0.1% FBS, which inhibited ALP expression when serving as a negative control. (**E**) Effect of serum deprivation in OGF-induced cells. Serum deprivation suppressed ALP expression. (**F**) Effect of ischemia. Ischemia was induced through cotreatment with 0.1% FBS and 200 µM CoCl_2_ to mimic nutrient and oxygen deprivation. Ischemia suppressed ALP expression and reduced cell density. (**G**) ALP activity. The O.D. value of the final products was normalized to the O.D. value of all of the cells. Each bar corresponds to a single group and represents the mean ± SD (*n* = 3). Asterisks indicate significant differences between two groups (* *p* < 0.05, ** *p* < 0.01, and *** *p* < 0.001). Abbreviations: ALP, alkaline phosphatase; Ctrl, control; FBS, fetal bovine serum; OGF, osteogenic growth factor.

**Figure 6 biomedicines-09-00685-f006:**
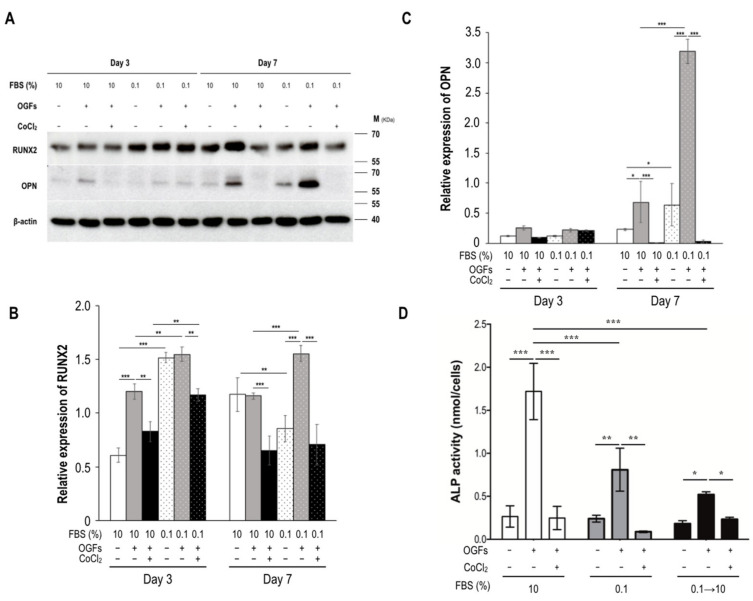
Expression patterns of osteoblast markers under the stress of serum insufficiency alone or in combination with hypoxia. (**A**) Immunoblotting was used to detect marker proteins in MC3T3-E1 cells after different treatments. Anti-RUNX2 and OPN antibodies were used to detect osteoblast markers. (**B**–**D**) All data are presented as mean ± SD (*n* = 3); asterisks at the tops of bars indicate significant differences (* *p* < 0.05, ** *p* < 0.01, *** *p* < 0.001). Quantification of (**B**) RUNX2 and (**C**) OPN expression levels. Identities of RUNX2 and OPN were normalized to that of the level of β-actin. (**D**) Serum reperfusion had no effect on ALP activity. Abbreviations: FBS, fetal bovine serum; M, marker; OGF, osteogenic growth factor; OPN, osteopontin; RUNX2, runt-related transcription factor 2.

**Figure 7 biomedicines-09-00685-f007:**
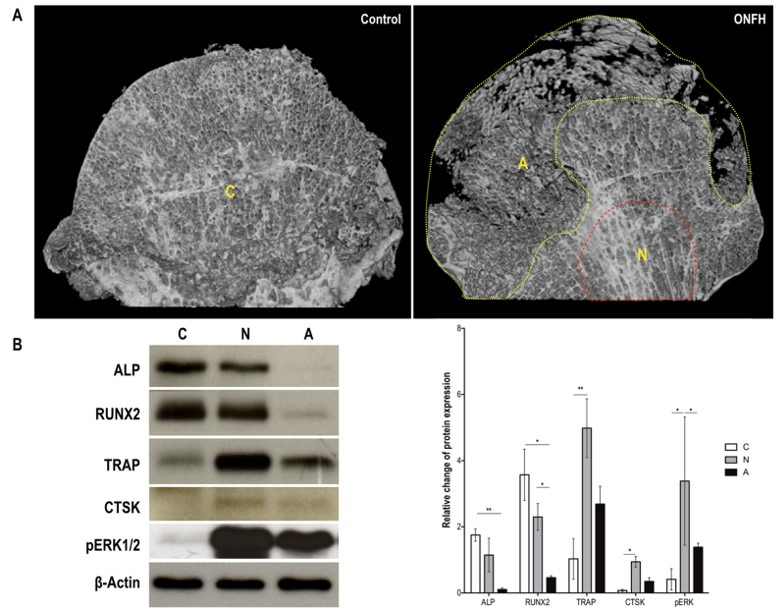
Expressions of osteoblast and osteoclast markers in the bone tissues of patients with ONFH. (**A**) Frontal section of µCT images of the femoral heads of patients in control and experimental (ONFH) groups. Letters indicate different tissue regions, namely, regions with normal bone (labeled as C) as well as non-necrotic (labeled as N) and necrotic (labeled as A) ONFH tissues for protein extraction. (**B**) Expression and quantification of osteoblast and osteoclast markers in the control and ONFH tissues. After proteins were extracted from the C, N (encompassed by the dashed red line near the femoral neck), and A (encompassed by the dashed yellow line) regions, osteoblast (ALP and RUNX2) and osteoclast (TRAP and CTSK) markers were used to detect the protein levels in the three regions. The density of the four bands was normalized to that of the internal control, β-actin. Each bar corresponds to a single group and represents the mean ± SD (control, *n* = 7; ONFH, *n* = 7). Asterisks at the tops of bars indicate significant differences (* *p* < 0.05 and ** *p* < 0.01) between groups. Abbreviations: A, avascular necrotic; ALP, alkaline phosphatase; CTSK, cathepsin K; ONFH, osteonecrosis of the femoral head; µCT, micro-computed tomography; RUNX2, runt-related transcription factor 2; TRAP, tartrate-resistant acid phosphatase.

## Data Availability

The data presented in this study are available on request from the corresponding author.

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
