# Peer review of "Serum Insufficiency Induces RANKL-Independent Osteoclast Formation during Developing Ischemic ONFH"

_biomedicines, 2021, doi:10.3390/biomedicines9060685_

Round 1

Reviewer 1 Report

A good and comprehensive workup. The presentation of the results is convincing, even if in some places the comprehensibility suffers due to bulky formulations. There are only minimal comments from my side: 

  • It would be desirable to consistently use corresponding B-actin bands in the graphical representation of the western blots to ensure comparability.Figure 3 c normalizes to ERK, perhaps I missed it in the text, otherwise a justification would be desirable why normalization to a classical house keeping gene is omitted here.
  • Overall, it is surprising that after scientific editing the language is still so unwieldy and partially incorrect. Is it perhaps a version before editing?

Apart from that, the scientific content is convincing and should be published.

Reviewer 2 Report

The authors of the paper “Serum insufficiency induces RANKL-independent osteoclast formation during developing ischemic ONFH” focused on the expression of osteoblasts and osteoclasts markers in vitro-murine macrophages, RAW264.7 cells-and ex vivo studies- osteonecrosis of the hip (ONFH)-during serum insufficiency and ischemic condition. Based on a variety of methods, the authors found that the co-treatment of serum restriction and hypoxia reduced alkaline phosphatase activity and mineralization in osteoblasts and increased tartrate-resistant acid phosphatase activity, phosphorylation of extracellular signal-regulated kinases and also ROS production. All together the results suggest the possible regulation of osteoclasts formation in ischemia-induced osteonecrosis by serum insufficiency.

 The subject addressed by the authors is interesting and relatively comprehensive presented.

However I have some comments to make:

-role of CoCl2 should be explained and also why the authors have used 2 concentrations in experiments

-For some experiments the behaviour/basal level of proteins/markers expression should be mentioned at least in the text

-Figure 4 g Calcification assay-what are the values for day 14?

-In all Figures:  lowercase in graph and uppercase in Caption!?

- English language not properly worded; terms and vocabulary style need to be improved; some parts of the text are written in a confusing manner and need to be rephrasing (i.e Abstract line 8 to 14!)

In conclusion, although the domain is important and the authors have done a huge work, the paper need to be improved before accepted for publication.

Round 2

Reviewer 2 Report

The new version of the paper has been improved compared to the initial one and this new form is accepted for publication. The answer of the authors covered all the problems mentioned by the reviewer.